# Elevating Smart Manufacturing with a Unified Predictive Maintenance Platform: The Synergy between Data Warehousing, Apache Spark, and Machine Learning

**DOI:** 10.3390/s24134237

**Published:** 2024-06-29

**Authors:** Naijing Su, Shifeng Huang, Chuanjun Su

**Affiliations:** 1Department of Project Management and Industrial Engineering, Shandong University, 27 Shanda Nanlu, Jinan 150100, China; 202313049@mail.sdu.edu.cn; 2Department of Industrial Engineering and Engineering Management, Yuan Ze University, 135, Far-East Rd., Taoyuan 320315, Taiwan; iesfhuang@saturn.yzu.edu.tw

**Keywords:** smart manufacturing, predictive maintenance, IOT, data warehousing, Apache Spark

## Abstract

The transition to smart manufacturing introduces heightened complexity in regard to the machinery and equipment used within modern collaborative manufacturing landscapes, presenting significant risks associated with equipment failures. The core ambition of smart manufacturing is to elevate automation through the integration of state-of-the-art technologies, including artificial intelligence (AI), the Internet of Things (IoT), machine-to-machine (M2M) communication, cloud technology, and expansive big data analytics. This technological evolution underscores the necessity for advanced predictive maintenance strategies that proactively detect equipment anomalies before they escalate into costly downtime. Addressing this need, our research presents an end-to-end platform that merges the organizational capabilities of data warehousing with the computational efficiency of Apache Spark. This system adeptly manages voluminous time-series sensor data, leverages big data analytics for the seamless creation of machine learning models, and utilizes an Apache Spark-powered engine for the instantaneous processing of streaming data for fault detection. This comprehensive platform exemplifies a significant leap forward in smart manufacturing, offering a proactive maintenance model that enhances operational reliability and sustainability in the digital manufacturing era.

## 1. Introduction

The aspiration of Industry 4.0 is to attain elevated automation by amalgamating cutting-edge technologies, such as the Internet of Things (IoT), machine connectivity, cloud computing, big data analytics, and artificial intelligence (AI) [1]. This integration demands a robust platform capable of optimizing the processes related to manufacturing, including predictive maintenance and quality assurance. Industry 4.0 represents the fourth industrial revolution, characterized by the interconnection of machinery, systems, and products, via digitalization and automation. This transformation is driven by the exponential growth in sensor technology, wireless communication, and advanced analytics, which collectively facilitate real-time data acquisition and processing.

Predictive maintenance (PDM) is particularly relevant in this context, as it leverages these advanced technologies to anticipate equipment failures before they occur, thus reducing downtime and maintenance costs. Traditional maintenance strategies, such as reactive maintenance (fixing equipment after failure) and preventive maintenance (scheduled maintenance regardless of equipment condition), are less efficient compared to PDM. By analyzing data from various sensors in real time, PDM allows for the timely detection of anomalies and the scheduling of maintenance activities based on the actual condition of the equipment. This not only enhances operational efficiency, but also extends the lifespan of machinery.

Moreover, the integration of IoT, AI, and big data analytics in manufacturing has revolutionized the way businesses operate. IoT enables the seamless connectivity of devices, allowing for continuous monitoring and data collection. AI algorithms can analyze this data to identify patterns and predict potential issues. Big data analytics provides the tools necessary to handle and process vast amounts of data, transforming it into actionable insights. Together, these technologies enable the development of intelligent, data-driven maintenance systems that improve reliability, safety, and productivity in manufacturing environments.

The vast amount of data generated by the IoT and contemporary manufacturing systems need to be converted into actionable insights. Analytics organizes this information and, using AI, helps businesses enhance their operations in several ways, as follows:Predictive maintenance: Sensors deliver data that signal potential equipment issues before failures occur, facilitating timely responses [2];Predictive demand: By evaluating both internal factors (like consumer preferences) and external elements (such as market trends), businesses can make precise forecasts, thus refining their product offerings;Elimination of bottlenecks: Big data helps identify performance inhibitors, allowing manufacturers to diagnose and resolve issues cost-effectively [3].

These are just a few of the advantages of big data analysis in manufacturing systems; other benefits include enhanced security, load optimization, and improved supply chain management [4,5]. To realize these benefits, a new generation of data-driven AIoT (artificial intelligence of things) platform is indispensable in the digital transformation process.

In this research, we focus on developing a data warehouse-driven platform for the predictive maintenance of production machinery in a smart manufacturing setting. The data-driven platform enables proactive maintenance by analyzing the condition of equipment in real time. Predictive maintenance (PDM) contrasts with preventive maintenance, which involves scheduled maintenance actions regardless of the equipment’s actual condition. PDM emphasizes monitoring equipment performance during normal operations to anticipate future failures, by analyzing data on variables such as vibration and temperature [6].

Big data analysis techniques can handle extensive data streams from multiple sources in real time, enabling PDM to maintain numerous devices, while reducing costs. These techniques continuously gather and scrutinize data using machine learning (ML) models to identify patterns for ongoing performance enhancement. The proposed platform includes the functions of automatic ML model selection and hyperparameter optimization, based on machine data and cross-validation results using random search [7].

Implementing PDM and other smart manufacturing applications requires three main components as follows, and illustrated in Figure 1:Data warehouse and ETL (extract, transform, load): This component extracts data from various sources, transforms it into a standard format, and loads it into a data warehouse, preparing it for machine learning and analysis;Big data analysis (BDA): This includes tools for machine learning and statistical modeling, allowing users to create predictive models with a GUI and apply them to solve real-world issues, thus improving decision-making and predicting future machine failures;Real-time data processing engine Apache Spark: Apache Spark [8] is a high-speed cluster computing engine designed for handling large volumes of streaming data, which effectively predicts machine faults and malfunctions in real time.

The core theme of modern smart manufacturing is data-driven decision-making based on key performance indicators (KPIs) and empirical data, rather than conjecture [9]. Predictive maintenance, a vital application in smart manufacturing, necessitates high-quality data. The proposed platform collects data via various sensors to monitor the machine conditions, stores it in a data warehouse using IoT technology, and generates ML models for real-time fault prediction.

The proposed predictive maintenance model is designed to be highly adaptable and generalizable across different types of manufacturing equipment and setups. By leveraging machine learning algorithms and big data analytics, the model can be trained on diverse datasets to identify patterns and predict failures in various equipment. For example, the model can be applied to both heavy machinery in automotive manufacturing and mixers and packaging machines in the food and beverage industry.

The DWPM architecture, which comprises (1) a data warehouse with ETL, (2) a big data Analysis (BDA) module, and (3) the real-time data processing engine Apache Spark, is designed to handle a wide range of manufacturing setups, ensuring its applicability across different environments. The platform includes several key components:

### Technological Developments in Smart Manufacturing

The evolution of smart manufacturing is driven by significant advancements in several key technologies, namely the Internet of Things (IoT), artificial intelligence (AI), and big data analytics. These technologies converge to create highly efficient, responsive, and intelligent manufacturing systems.

Internet of Things (IoT): IoT plays a critical role in smart manufacturing by enabling the interconnection of various devices and systems. IoT devices, equipped with sensors and communication modules, collect and transmit data from manufacturing processes in real time. This data includes information on machine performance, environmental conditions, and product quality. The continuous flow of data allows for real-time monitoring and decision-making, which is essential for predictive maintenance and other smart manufacturing applications.

Artificial intelligence (AI): AI technologies, including machine learning and deep learning, are at the forefront of analyzing the vast amounts of data generated by IoT devices. AI algorithms can process complex datasets to identify patterns, predict outcomes, and optimize processes. In predictive maintenance, AI is used to analyze sensor data to detect anomalies that indicate potential equipment failures. This enables maintenance to be performed proactively, reducing downtime, and extending the equipment lifespan.

Big data analytics: The ability to handle and analyze large volumes of data is a cornerstone of smart manufacturing. Big data analytics involves the use of advanced tools and techniques to extract meaningful insights from data. This includes data mining, statistical analysis, and visualization. In the context of predictive maintenance, big data analytics processes data from various sources to provide a comprehensive view of equipment health. This allows manufacturers to make informed decisions and implement maintenance strategies that are both efficient and cost effective.

Integration of technologies: The integration of IoT, AI, and big data analytics forms the backbone of modern smart manufacturing systems. IoT devices collect data, which is then processed and analyzed using AI and big data tools. The results of these analyses are used to optimize manufacturing processes, improve product quality, and enhance overall efficiency. For example, real-time data processing engines like Apache Spark can handle large streams of data, providing immediate insights and enabling quick responses to potential issues.

The synergy between these technologies is exemplified in our proposed platform, which combines data warehousing, Apache Spark, and machine learning to create a unified predictive maintenance system. This platform leverages the strengths of each technology to provide a comprehensive solution for maintaining and optimizing manufacturing equipment. By integrating these advanced technologies, the platform not only enhances the predictive maintenance capabilities, but also supports the broader goals of Industry 4.0, driving the transition towards more intelligent and autonomous manufacturing systems.

## 2. Related Works

This chapter provides an overview of the technologies pertinent to this study. Section 2.1 covers data warehousing technology with a focus on the extract, transform, load (ETL) processes. Section 2.2 explores the application of Apache Spark for real-time data processing. Section 2.3 discusses machine learning algorithms as the inference engine for PDM. Finally, Section 2.4 addresses the recent advancements in Industry 5.0.

### 2.1. Data Warehouses

In recent years, the demand for extensive information has significantly increased, elevating the importance of data warehouses. They have become crucial components of strategic planning and foundational elements of decision support infrastructures in numerous companies. The concept of data warehousing emerged in the 1980s as a solution to the limited availability of information provided by online application systems. These systems were praised by a narrow user base and lacked integration [10]. Devlin and Murphy were the pioneers who introduced the concept of data warehousing [11]. They proposed the use of read-only databases to store historical operational data. Subsequently, Inmon defined a data warehouse as a subject-oriented, integrated, time-variant, and non-volatile collection of data, designed to support the management decision-making process [12].

Traditional databases designed for transactional processing are not optimized for analytics. Conversely, a data warehouse is structured to facilitate efficient and rapid analytical processing [13]. As illustrated in Figure 2, the structure of a data warehouse is distinct from that of independent or dependent data marts developed from traditional databases.

Figure 3 depicts a generalized view of data warehouse architecture applicable across various real-life applications. Each data warehousing application involves the extraction of data from key systems using minimal resources, the transformation of that data based on specific rules from source to target, and loading the processed data into the data warehouse. This process is commonly known as the ETL (extract, transform, load) process [14].

Leveraging the architecture illustrated in Figure 3, data warehouses have been utilized across various business sectors, including social media platforms [15], the manufacturing industry [16], marketing [17], banking [18], education [19], and finance [20].

### 2.2. Apache Spark for Real-Time Data Processing

Handling vast quantities of data frequently necessitates parallelization and cluster computing, for which Apache Spark (Spark) has become the industry standard. Developed by the Algorithms, Machines, and People Lab at the University of California, Berkeley, Spark [21] is an open-source, general-purpose distributed computing system designed for big data analytics. Spark significantly outperforms earlier big data tools, such as Apache Hadoop, due to its in-memory caching and optimized query execution. It offers dedicated libraries for machine learning (Spark MLlib), stream processing (Spark Streaming), and SQL queries (Spark SQL) [22].

Spark excels in real-time data processing through its robust in-memory computation capabilities, enabling it to handle and process data streams in real time. Spark Streaming, a core component, supports scalable, high-throughput, fault-tolerant stream processing of live data streams. This allows for near-instantaneous data processing, making it suitable for applications that require real-time analytics and immediate feedback. Spark achieves this through a micro-batch processing model, where incoming data is divided into small batches for processing.

In addition to the main computing framework, Spark includes various libraries, such as Spark Streaming, Spark MLlib, Spark SQL, and Spark GraphX [23], as illustrated in Figure 4.

### 2.3. Machine Learning Algorithms for Predictive Maintenance

The application of machine learning (ML) algorithms in predictive maintenance has garnered significant attention in recent years, owing to their ability to preemptively identify potential failures and optimize maintenance schedules. Predictive maintenance (PDM) aims to predict equipment failures before they occur, thereby minimizing downtime and reducing maintenance costs.

Various machine learning algorithms have been employed for predictive maintenance, each with its own unique advantages and application scenarios. Commonly used algorithms include decision trees, support vector machines (SVMs), neural networks, and ensemble methods.

Decision trees are widely used for their simplicity and interpretability. They work by splitting the data into subsets based on the value of the input features, creating a tree-like model of decisions. Random forests, an ensemble method based on decision trees, enhance predictive accuracy by combining the predictions of multiple decision trees [24]. They are particularly effective in handling large datasets with numerous features.

Support vector machines are powerful classifiers that work by finding the optimal hyperplane that separates data points of different classes. SVMs are effective in high-dimensional spaces and are robust in regard to overfitting, particularly in scenarios where the number of dimensions exceeds the number of samples [25]. Their application in PDM has been demonstrated in various studies, showing high accuracy in fault detection and classification tasks.

Neural networks, particularly deep learning models, have shown exceptional performance in predictive maintenance due to their ability to model complex, non-linear relationships in data. Convolutional neural networks (CNNs) and recurrent neural networks (RNNs) are commonly used architectures. CNNs are well suited for processing grid-like data, such as images or sensor grids, making them ideal for visual inspection tasks [26]. RNNs, on the other hand, excel in handling sequential data, making them suitable for time-series analysis and anomaly detection in sensor data [27].

Ensemble methods, such as boosting and bagging, combine the predictions of multiple base learners to improve the overall predictive performance. Techniques like gradient boosting machines (GBMs) and extreme gradient boosting (XGBoost) have been successfully applied to predictive maintenance tasks, offering high accuracy and robustness [28]. These methods are particularly advantageous in scenarios with imbalanced datasets, a common challenge in PDM applications.

In the manufacturing sector, ML algorithms have been integrated into industrial IoT systems to monitor equipment health and predict failures in real time. Recent advancements have shown that integrating IoT, AI, and big data analytics can significantly enhance the accuracy and efficiency of predictive maintenance systems [29]. Machine learning techniques are widely applied in predictive maintenance to predict pending failures in advance and address challenges, such as failure forecast, anomaly detection, and remaining useful life prediction. Additionally, predictive maintenance machine learning algorithms are experimentally applied in the design of IoT-based condition monitoring systems [30].

The integration of machine learning algorithms in mechatronic systems for predictive maintenance offers the ability to analyze data, identify patterns, and predict potential failures in advance, enhancing the efficiency and reliability of maintenance strategies. Furthermore, machine learning algorithms, specifically a CNN–LSTM network, are utilized for predictive maintenance functions to enhance cost efficiency and machine availability in manufacturing plants [31].

Supervised AutoGluon obtained the best results for all machine learning tasks in the study on predictive maintenance, including predicting the number of days until the next equipment failure [32].

Machine learning algorithms, such as random forests, gradient boosting, and deep learning, were utilized in a case study on predictive maintenance in manufacturing management, showcasing improved accuracy in predicting equipment failures and potential cost reductions [33]. Optimizing predictive maintenance with machine learning significantly improves equipment reliability, overcoming the limitations of conventional methods [34].

### 2.4. Recent Advancements in Industry 5.0

Industry 5.0 represents the next evolution in industrial development, building upon the foundations of Industry 4.0, which emphasized the integration of cyber-physical systems, the Internet of Things (IoT), and big data analytics, to create smart manufacturing environments [35]. Industry 5.0 introduces a more human-centric approach, where human intelligence and creativity are integrated into the automation and digitalization processes. This new paradigm aims to enhance resilience, flexibility, and sustainability in manufacturing systems by leveraging advanced technologies, while ensuring that human workers remain an integral part of the production loop.

#### Human Workers in the Loop for Enhanced Resilience and Intelligence

One of the core tenets of Industry 5.0 is the concept of “human-in-the-loop” (HITL) systems, where human workers collaborate with automated systems to enhance decision-making and problem-solving capabilities. This approach not only preserves the role of human expertise and intuition, but also enhances the adaptability and resilience of manufacturing systems. Key aspects of integrating human workers into smart manufacturing systems include real-time data analytics, intelligent decision-making, and collaborative automation [36].

## 3. Methodology and System Architecture

In a busy production environment, identifying abnormalities often occurs too late, resulting in numerous rejected products, increased production costs, and delays to scheduled deliveries. These abnormalities typically stem from production equipment failures. In a smart manufacturing context, signals such as vibrations, torque, temperature, and humidity can be collected from production equipment via machine outputs or sensors. Predictive maintenance leverages these signals and machine learning technology to monitor and predict equipment failures in real time. Our proposed data warehouse-driven real-time predictive maintenance (DWPM) platform for smart manufacturing is designed to handle the entire data lifecycle, from data extraction and processing to analysis and real-time decision-making. This section describes the end-to-end architecture of the platform, highlighting the extract, transform, load (ETL) processes, data warehousing, big data analysis, and real-time data processing components.
ETL Processes

The ETL component is crucial for collecting sensor data from industrial and production equipment. The data undergoes preprocessing steps, including extraction, out-of-bounds cleaning, and treatment of missing values. Our approach employs a chain of filters to manage the data effectively.

Data extraction: Data is extracted from various sources, including sensors, IoT devices, and log files;Data transformation: The extracted data is cleaned and transformed into a standardized format suitable for analysis. This includes handling missing values, outlier detection, and normalization;Data loading: The transformed data is loaded into a data warehouse, ensuring it is stored efficiently and is easily accessible for analysis.

The proposed DWPM architecture consists of three modules, as depicted in Figure 5: (1) a data warehouse (DW), (2) big data analysis (BDA), and (3) the real-time data processing engine, Spark (RDPS). Each module is composed of two components. the DW module includes extract, transform, and load (ETL) processes, and data storage. The BDA module comprises feature engineering and model generation. Lastly, the RDPS module incorporates model deployment for inference and prediction, and a warning system. 

### 3.1. Data Warehouse (DW) Module

The necessity for data warehouses has escalated due to the growing demand for deriving actionable insights from data. Rather than operating multiple predictive maintenance environments independently, which can lead to conflicting information, a data warehouse should consolidate all information sources [37].

In this module, the ETL component is crucial for collecting sensor data from industrial and production equipment. This data undergoes preprocessing steps, including extraction, out-of-bounds cleaning, and treatment of missing values [38].

#### 3.1.1. Missing Data Treatment

Effective data preprocessing is essential for ensuring the reliability and accuracy of our DWPM. In this section, we detail the methods used for handling missing data, which is a common issue in real-time data streams from industrial sensors. We focus on imputation, interpolation, and the use of machine learning techniques for estimating missing values in real time [39].

The theoretical foundation for these methods is rooted in statistical inference and machine learning principles. Imputation methods rely on the statistical properties of the data, such as the central tendency and variance. Interpolation methods use mathematical functions to estimate intermediate values, ensuring smooth transitions between observed data points [40]. Machine learning techniques leverage the underlying patterns and structures in the data to make informed predictions about missing values. These methods are integrated into the data preprocessing pipeline of DWPM. The steps are as follows:Data collection: Sensor data is collected in real-time and stored in a NoSQL database (MongoDB) designed for high throughput and scalability;Data cleaning: Raw data undergoes initial cleaning to remove outliers and standardize formats. Missing values are identified during this stage;Missing data handling: Depending on the nature and extent of missing data, appropriate methods (imputation, interpolation, or machine learning) are applied as follows:
For small, sporadic gaps, mean/median imputation or linear interpolation is used [41];For larger or systematic gaps, KNN imputation or spline interpolation is applied [42];For complex and high-dimensional data, random forest imputation, autoencoders, or Gaussian processes are used [43].
Data loading: Processed data, now complete and consistent, is loaded into the data warehouse, ready for further analysis and model training.

By employing these methods, we ensure that the data used in our predictive maintenance model is as accurate and reliable as possible, thereby enhancing the robustness of real-time data processing and the effectiveness of predictive maintenance interventions.

#### 3.1.2. MongoDB-based Data Warehouse Cluster

NoSQL databases, or “not only SQL” databases, differ from conventional tabular storage by offering various data models, including document, key-value, wide-column, and graph databases. These databases feature flexible schemas and can scale efficiently to manage large data volumes and high user loads [44]. The increasing demand for predictive maintenance analytics necessitates the near real-time loading of data streams into the data warehouse. Due to the schema-less and sometimes undefined formats of data streams from multiple sources, a document-oriented real-time NoSQL data warehouse is preferred over a traditional relational DBMS-based data warehouse.

MongoDB, a NoSQL database adopted in our implementation, stores collections of schema-less documents. Each document, defined by a group of key/value pairs, operates with a dynamic schema, making MongoDB a high-performance data structure suitable for lookup and filtering operations [45]. High availability in the data warehouse cluster is achieved through load balancing and redundancy, ensuring uninterrupted access, even if a server fails. MongoDB supports two types of clustering: replica sets and sharding. For this study, sharding is employed to ensure load balancing and high availability by distributing data across multiple machines, as illustrated in Figure 6.

### 3.2. Big Data Analysis (BDA) Module

The BDA module consists of two primary components: feature engineering and machine learning. Feature engineering is a crucial phase in the predictive maintenance process, where preprocessed data from a data warehouse is transformed and refined to create significant input features for a machine learning model. This involves selecting pertinent variables, generating new features from raw data, and modifying existing features to better represent the underlying patterns and trends. For predictive maintenance, this includes generating features such as historical equipment usage patterns, the mean and variance of sensor readings, and anomaly detection scores.

In this research, we employ MongoDB’s adaptable document model and aggregation framework to store features such as temperature, vibration, humidity, pressure, and acoustic data within a feature store. The feature store functions as a centralized repository designed for storing, sharing, and managing machine learning features.

In the machine learning process, equipment sensor data stored in the data warehouse is utilized to train models. The big data analysis (BDA) platform, built on top of the H_2_O open-source machine learning framework [46], leverages the powerful capabilities of Spark, an open-source, in-memory computing platform known for its efficiency and versatility. Integrating the BDA platform with Spark allows users to seamlessly connect devices via Spark Streaming, process the data, and build predictive models using the BDA platform’s ML Libraries. The BDA platform offers a suite of algorithms for training machine learning models on large datasets and performing related statistical analyses. The process of making predictions with the BDA platform involves three steps: initializing the model, fitting it to the training data, and predicting new values. The output is an array of data corresponding to the number of equipment devices in the test set, indicating whether each device is likely to fail. The BDA platform architecture is illustrated in Figure 7.

#### 3.2.1. Machine Learning Modeling

##### Dataset Partitioning Strategy

Effective dataset partitioning is crucial for building robust machine learning models. In this study, we employed a three-way partitioning strategy to divide the dataset into training (70%), validation (15%), and test (15%) sets. This approach ensures that the model is trained on a substantial portion of the data, validated on a separate subset to fine tune the hyperparameters, and tested on an independent set to evaluate its performance.

##### Model Training Process

The model training process involves selecting appropriate algorithms, applying cross-validation techniques, and fine tuning the hyperparameters to achieve optimal performance. We adopted a random forest model for predicting machine failures using sensor data, which offers several advantages, especially in the context of industrial applications where reliability and accuracy are critical. The random forest model’s combination of accuracy, robustness, scalability, and flexibility makes it an excellent choice for predicting machine failures using sensor data. It not only improves prediction performance, but also provides valuable insights into the factors influencing machine health, aiding in effective predictive maintenance strategies. We also employed 10-fold cross-validation to ensure robust evaluation of the model’s performance.

##### Hyperparameter Tuning

To optimize the random forest model, we used random search for hyperparameter tuning. Random search explores a wide range of hyperparameter values by sampling randomly, which is computationally efficient. For more complex optimizations, we considered grid search, which exhaustively searches through a specified hyperparameter space, and Bayesian optimization techniques to find the optimal set of hyperparameters.

##### Training Results

The performance of the random forest model was evaluated using several key metrics, including accuracy, precision, recall, and F1 score. These metrics provide a comprehensive view of the model’s performance across different aspects of prediction.

### 3.3. Real-Time Data Processing Engine, Spark

The real-time data processing engine, Spark, is integral to the DWPM, as depicted in Figure 8. This engine facilitates the seamless collection, storage, and analysis of sensor data in real time. The process begins with the collection of data from various sensors, measuring parameters such as temperature, speed, torque, and tensile strength. The captured data is streamed with Apache Bahir and the Spark Streaming MQTT Connector to enable the ingestion of the streaming data into the system [47].

Once collected, the data is processed in batches by Spark’s real-time streaming capabilities, transforming raw input into structured data. The Spark Executor then plays a critical role in processing this data, executing tasks that prepare it for storage. The processed data is stored in a data warehouse cluster, ensuring it is readily available for subsequent analysis.

For the analysis phase, the MongoDB Connector facilitates the transfer of data from the data warehouse to the analysis engine [48]. This engine utilizes advanced analytics libraries with H_2_O.ai, to perform real-time predictions and generate warnings. The integration of these components ensures that predictive maintenance tasks can be performed efficiently, allowing for the timely detection of potential equipment failures and enabling proactive maintenance interventions.

By leveraging Apache Spark’s robust data processing capabilities, the DWPM can handle large volumes of streaming data with high throughput and low latency, thus enhancing the reliability and efficiency of maintenance operations.

#### Ensuring Minimal Latency in Fault Detection

To ensure minimal latency in fault detection when handling high-throughput data streams, our platform leverages the following strategies and optimizations within Apache Spark:High-throughput ingestion: The platform uses Apache Bahir as a high-throughput message broker to collect and buffer data from IoT sensors. Spark Streaming reads data from Bahir in micro-batches, ensuring efficient data ingestion and processing;Parallel processing: Spark’s distributed architecture allows the platform to process data in parallel across multiple nodes. This parallel processing capability reduces the time required to analyze large volumes of data, enabling quicker fault detection;Sliding windows and aggregations: The platform uses sliding windows and time-based aggregations to continuously monitor and analyze data streams. These operations allow the system to detect trends, anomalies, and patterns indicative of potential equipment faults;Threshold-based alerts: The platform employs threshold-based alerting mechanisms to trigger immediate notifications when certain metrics exceed predefined limits.

By leveraging Apache Spark’s real-time data processing capabilities and implementing these strategies, our platform ensures minimal latency in fault detection, while handling high-throughput data streams.

### 3.4. Data Privacy and Security

To ensure data privacy and security, we have implemented several measures, as follows:Encryption: All data transmitted between IoT devices and the cloud are encrypted using industry standard protocols, such as TLS;Access controls: Strict access controls are in place to ensure that only authorized personnel can access sensitive data;Compliance: The platform complies with relevant data privacy and security standards, including the GDPR and ISO/IEC 27001.

## 4. DWPM Implementation

To validate the feasibility of the proposed DWPM, a prototype was developed to fully demonstrate its usability and functional design. The DWPM implementation aims to meet three primary requirements: (1) scalability to handle petabytes of data; (2) support for low-latency data access and decision-making; and (3) an integrated analytics environment to expedite advanced analytics modeling and operationalization processes. This section details the implementation of three core components: (1) the real-time data processing engine, Spark; (2) the data warehouse; and (3) the big data analysis module.

### 4.1. Spark Standalone Cluster

To implement the data warehouse-driven real-time predictive maintenance (DWPM) platform, 17 virtual machines were established on Cisco Hyperconverged Infrastructure, all running Linux operating systems. Real-time data from sensors, IoT devices, and log files require constant monitoring and immediate processing. Consequently, a highly scalable, reliable, and fault-tolerant data streaming engine is essential for real-time data analysis. 

Spark Streaming, a core component of Spark’s API, is utilized for real-time data processing. It supports processing data from various input sources and storing the processed data into multiple output sinks. Apache Bahir, an open-source project, extends distributed analytics platforms like Spark, providing plugins such as streaming-mqtt for remote sensing and equipment monitoring in challenging environments. This research implements Spark Streaming and Apache Bahir for streaming data, as illustrated in Figure 9. 

Spark can operate on several existing cluster managers, including standalone, Apache Mesos, and Hadoop YARN [49]. The standalone mode, a simple cluster manager integrated with Spark, is easy to set up and can run on Linux, Windows, or Mac OSX. It is the simplest way to run Spark applications in a clustered environment.

The standalone cluster mode features a master–slave architecture, consisting of master and slave processes to run Spark applications. The master acts as the cluster’s resource manager, accepting applications and scheduling resources to run them. Workers are responsible for launching executors for task execution. After installing the master and worker nodes, the Spark Web UI can be accessed for monitoring (https://localhosts.mobi/9000), accessed on 10 May 2024, as shown in Figure 10.

### 4.2. Data Warehouse

There are two main components to be built in the data warehouse, one is the ETL system Apache NiFi for data extraction and data cleaning, and the other is the data warehouse system with MongoDB cluster implementation for storing subject-oriented data. 

#### 4.2.1. Apache NiFi

Apache NiFi [50] is used for its robust data extraction and cleaning capabilities. It facilitates the ingestion of data from various sources, ensures data integrity, and prepares the data for storage in the data warehouse, as shown in Figure 11. NiFi’s user-friendly interface and flexible data routing allow for efficient and scalable ETL processes, which are critical for maintaining high-quality data in the DWPM platform.

#### 4.2.2. MongoDB Cluster

MongoDB offers robust support for handling large volumes of data, high concurrency, and reliability, surpassing traditional relational databases. Compared to other NoSQL databases, MongoDB’s document-based data model and dynamic schema make it more flexible. Its sharding architecture allows for horizontal scaling to accommodate vast amounts of data. 

In MongoDB, a sharded cluster comprises, as follows:Shards: Replica sets containing subsets of the cluster’s data;Mongos: Instances that act as an interface between client applications and the sharded cluster, routing queries and writing operations for the appropriate shards;Config servers: These maintain the sharding metadata and are the authoritative source of cluster configuration.

For instance, the MongoDB cluster system architecture, depicted in Figure 7, includes six shards (divided into two clusters), one mongo, and three config servers. Integrating this setup with Apache NiFi and a Spark standalone cluster extends the system architecture, as shown in Figure 9, with the results illustrated in Figure 12.

##### Warehouse

Apache NiFi automates data capture from data sources, follows predefined data cleansing workflows, and stores the processed data in the data warehouse. The MongoDB Connector for Apache Spark leverages MongoDB’s aggregation pipeline and secondary indexes to efficiently extract, filter, and process required data ranges. To optimize performance across large, distributed datasets, this Connector co-locates resilient distributed datasets (RDDs) with the source MongoDB nodes, reducing data movement across the cluster and minimizing latency.

### 4.3. Big Data Analysis (BDA) Module

Dell Technologies’ third Digital Transformation Index survey in 2020 identified the inability to extract valuable insights from data and/or data overload as the third most significant barrier to digital transformation [52]. This paradox highlights that while data is a company’s most crucial asset, it can also hinder transformation efforts. Companies require extensive data to address and solve questions, yet the current data collection rate surpasses their ability to analyze and apply it effectively. The BDA module is designed to tackle this challenge by focusing on rapid model development, swift model retraining, and streamlined or automated analysis processes.

To meet these objectives, we utilize Sparkling Water, developed by H_2_O.ai, as our big data analysis platform. H_2_O is a prominent open-source machine learning and AI platform that includes widely used machine learning algorithms, such as generalized linear modeling, random forest, and gradient boosting machine (GBM). Sparkling Water integrates H_2_O’s fast, scalable machine learning algorithms with Apache Spark’s capabilities, allowing users to drive computation from Scala, R, or Python, while leveraging the H_2_O Flow UI. This integration provides an optimal machine learning platform for application developers. The hardware and software configurations for Sparkling Water are summarized in Table 1.

With the implementation of the DWPM platform completed, the overall system infrastructure is illustrated in Figure 13, extending the foundation shown in Figure 13.

### 4.4. Practical Results and Application Cases

The practical application of the DWPM platform was demonstrated using real-world data from a leading manufacturer of polarizer films. The data encompassed two production lines, referred to as Line A and Line B, each with 15 monitoring points, totaling approximately 4 million data points collected from 2016 to 2017. The demonstration was divided into three stages: data acquisition automation, model retraining, and deployment of newly trained models.
Stage 1: Data Acquisition Automation

Using Apache NiFi, data acquisition was automated with a web-based user interface developed using Node.js. This interface connected to a relational database (MariaDB) to ensure that data was correctly captured, cleaned, and transferred to the NoSQL data warehouse (MongoDB), as shown in Figure 14. The successful automation of the data acquisition ensured the seamless flow of data, which is critical for real-time predictive maintenance.
Stage 2: Model Retraining

Model retraining was performed to maintain the accuracy and responsiveness of the predictive models in the face of data and concept drift. The platform supported both automated and manual retraining. For this demonstration, manual retraining was showcased, highlighting the platform’s capability to adapt and improve its predictive models continuously, as shown in Figure 15. New models were trained using H_2_O.ai’s Flow interface, ensuring high accuracy and reliable maintenance predictions.
Stage 3: Deployment of Newly Trained Models

The deployment of newly trained models involves a two-step approach to enhance the original monitoring system. This comprises: (1) the model deployment and prediction component, a background service at the equipment level that streams data to the analysis platform, and (2) the warning system component on the analysis server, which receives and analyzes the streaming data, reporting device conditions to users.

The system filters machines that have completed model building and indicates model accuracy. Upon user confirmation, the model is deployed on the DWPM platform. For real-time alerting, the model management and real-time data monitoring interface lists all the models. It identifies production lines and machines with deployed models, describes the service status, and manages the data streaming settings. Accurate data sources are crucial for maintaining model accuracy. The predictive results indicate the machine state, namely abnormal (requiring maintenance), not connected, or normal under analysis, as shown in Figure 16. The real-time data presentation highlights the status of the selected machine.

#### Effectiveness of the Platform

The practical application of the DWPM platform demonstrated significant improvements in the predictive maintenance capabilities. Automated data acquisition and processing ensured that real-time data was always up to date and accurate. The retraining and deployment of predictive models allowed for the continuous improvement of maintenance predictions, reducing downtime and maintenance costs.

## 5. Conclusions

This study successfully demonstrated the development and implementation of a data warehouse-driven real-time predictive maintenance (DWPM) platform, showcasing its feasibility and functional design. The platform integrates three core components, a data warehouse, a big data analysis (BDA) module, and the real-time data processing engine, Apache Spark.

The DWPM platform addresses key requirements, essential for modern smart manufacturing environments, namely scalability to handle petabytes of data, support for low-latency data access and decision-making, and an integrated analytics environment to expedite advanced analytics modeling and operationalization processes. The implementation leveraged the robust capabilities of Apache NiFi for data extraction and cleaning, MongoDB for scalable data storage, and Apache Spark for real-time data processing, ensuring high performance and reliability.

Overall, the DWPM platform exemplifies a significant advancement in smart manufacturing, offering a proactive maintenance model that enhances operational reliability and sustainability. The successful implementation of this platform demonstrates its potential to transform data-driven decision-making processes in the manufacturing industry, paving the way for future innovations in predictive maintenance and smart manufacturing technologies.

## Figures and Tables

**Figure 1 sensors-24-04237-f001:**
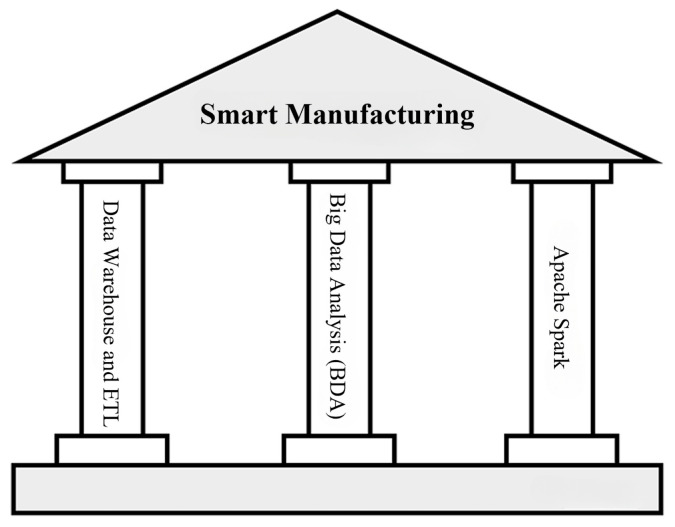
Three key components for realizing the benefits of smart manufacturing.

**Figure 2 sensors-24-04237-f002:**
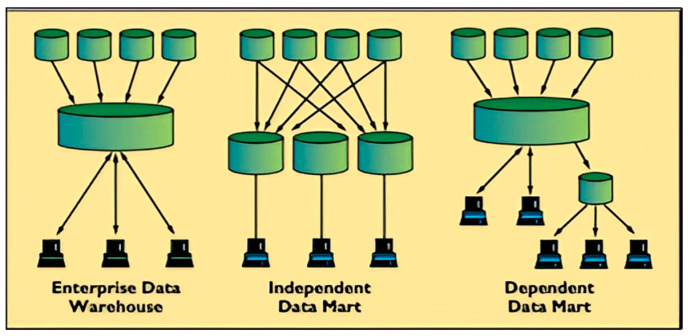
The differences between a data warehouse and data marts.

**Figure 3 sensors-24-04237-f003:**
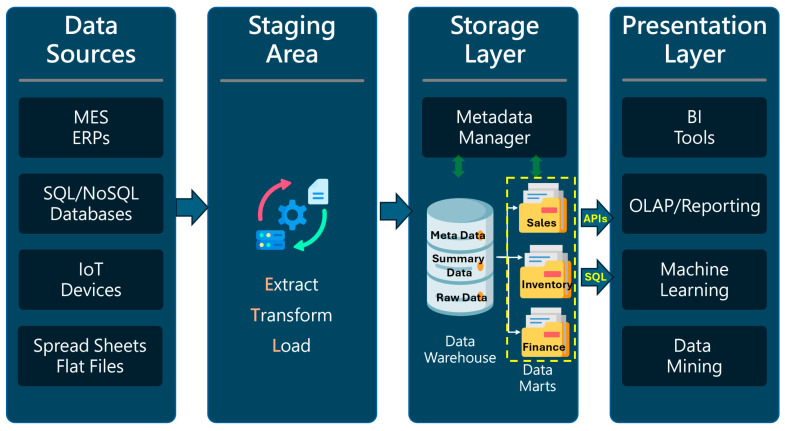
The general architecture of a data warehouse.

**Figure 4 sensors-24-04237-f004:**
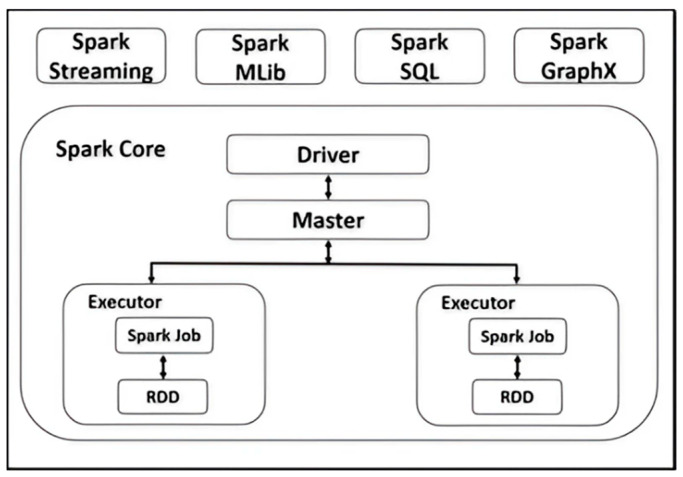
The architecture of Spark.

**Figure 5 sensors-24-04237-f005:**
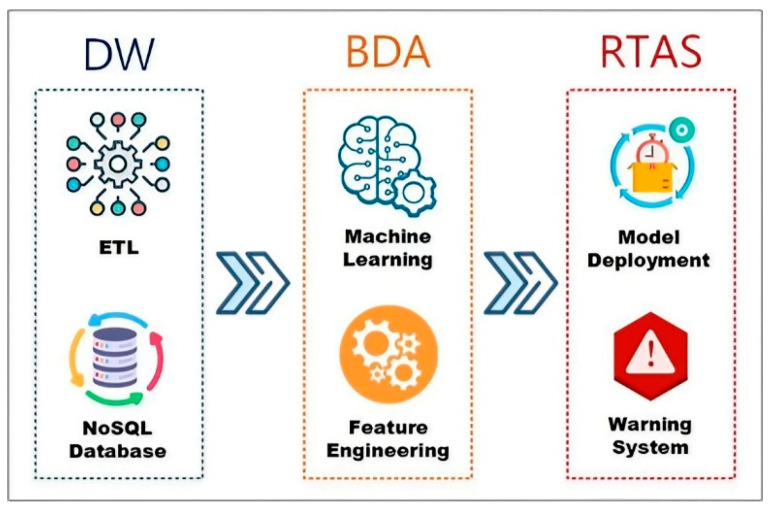
DWPM architecture.

**Figure 6 sensors-24-04237-f006:**
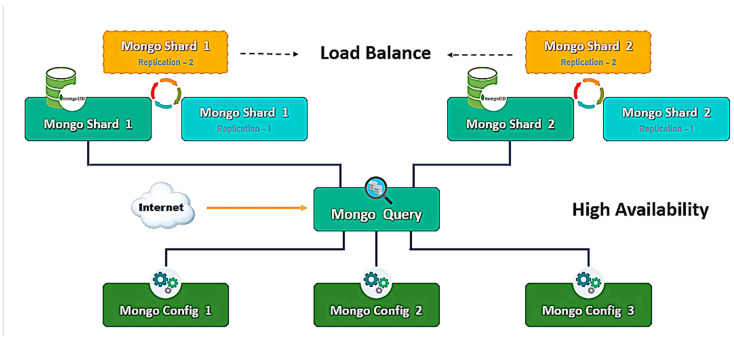
Data warehouse cluster architecture.

**Figure 7 sensors-24-04237-f007:**
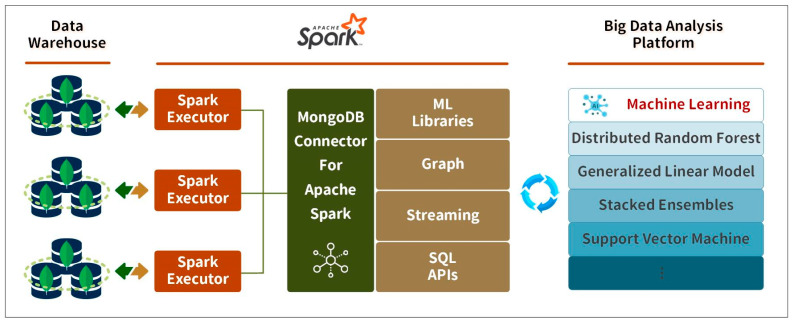
Big data analysis platform.

**Figure 8 sensors-24-04237-f008:**
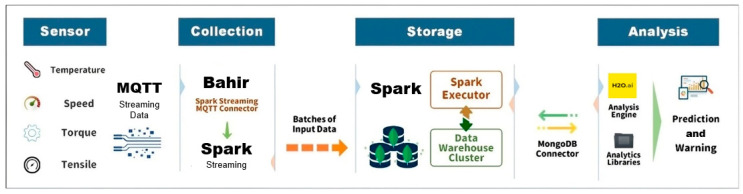
DWPM infrastructure.

**Figure 9 sensors-24-04237-f009:**
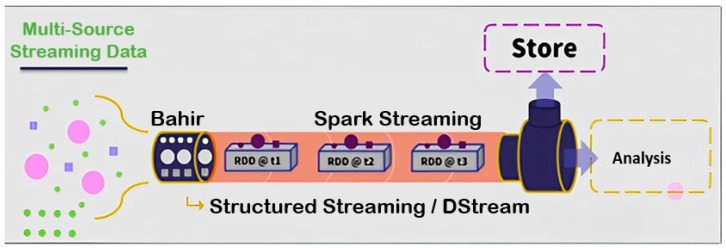
Spark Streaming uses DStream to transform streaming data into a series of batches.

**Figure 10 sensors-24-04237-f010:**
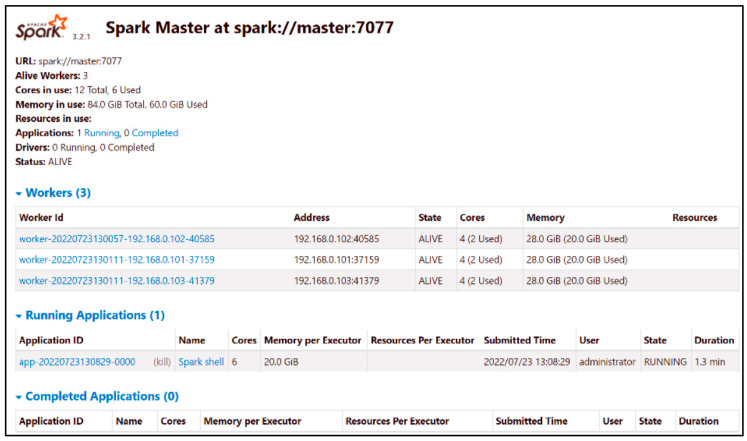
Master node web UI.

**Figure 11 sensors-24-04237-f011:**
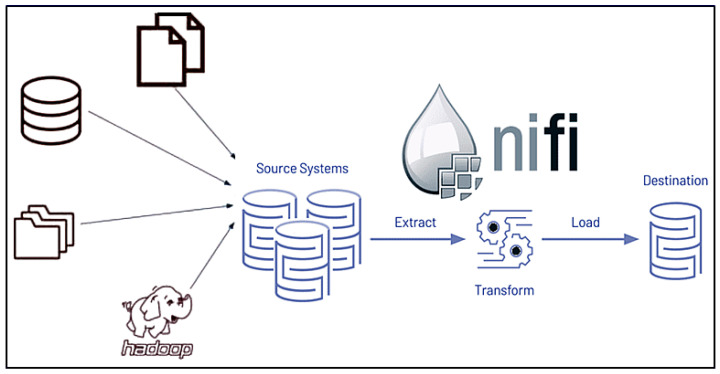
NiFi ETL [51].

**Figure 12 sensors-24-04237-f012:**
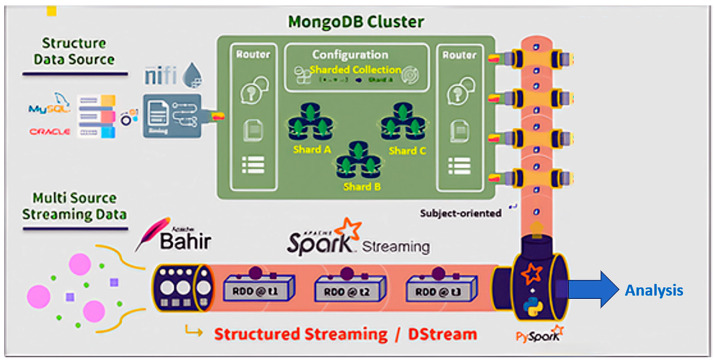
Structured data and streaming data integration.

**Figure 13 sensors-24-04237-f013:**
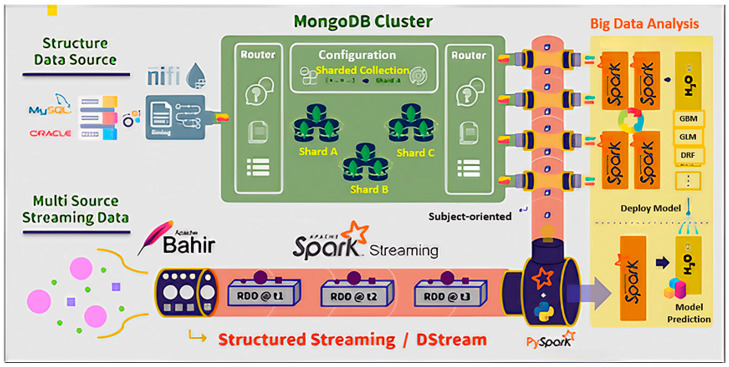
DWPM infrastructure.

**Figure 14 sensors-24-04237-f014:**
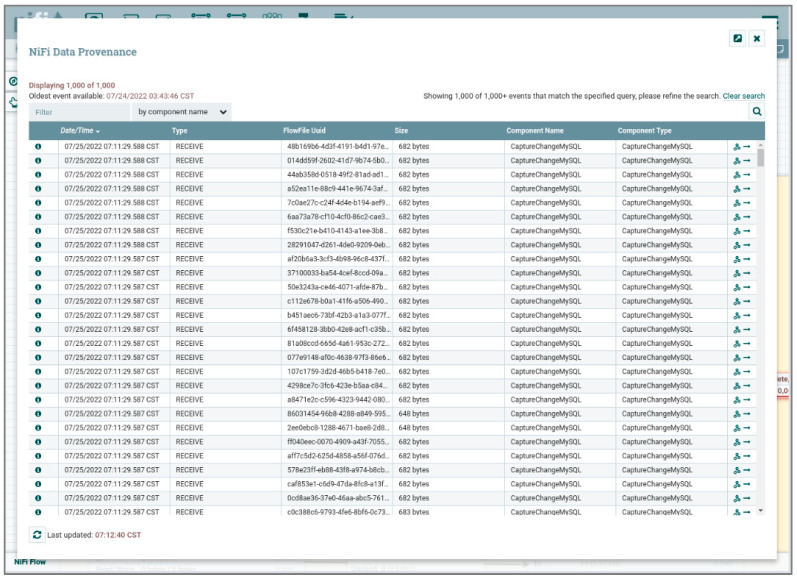
Data acquisition and data cleaning by NiFi.

**Figure 15 sensors-24-04237-f015:**
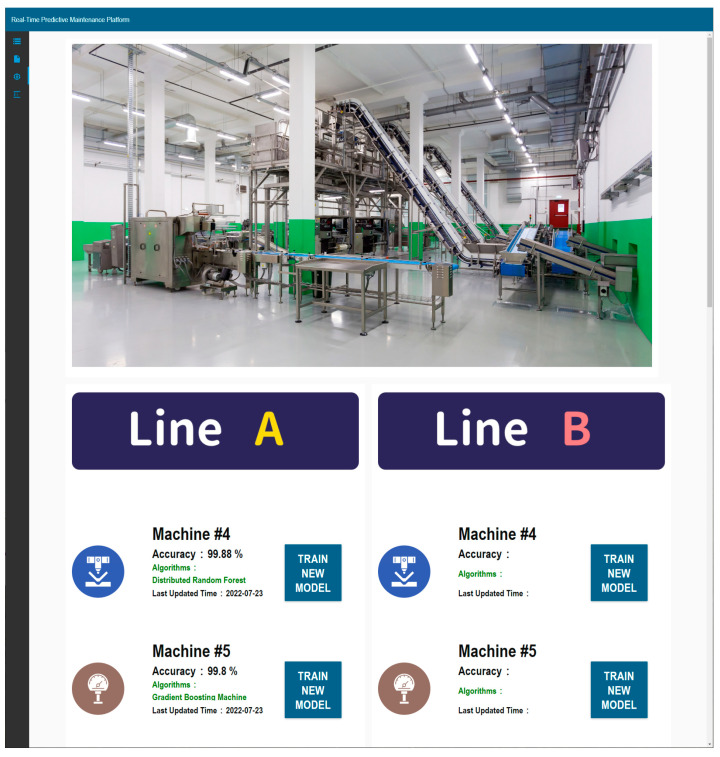
Model retraining page on the DWPM platform.

**Figure 16 sensors-24-04237-f016:**
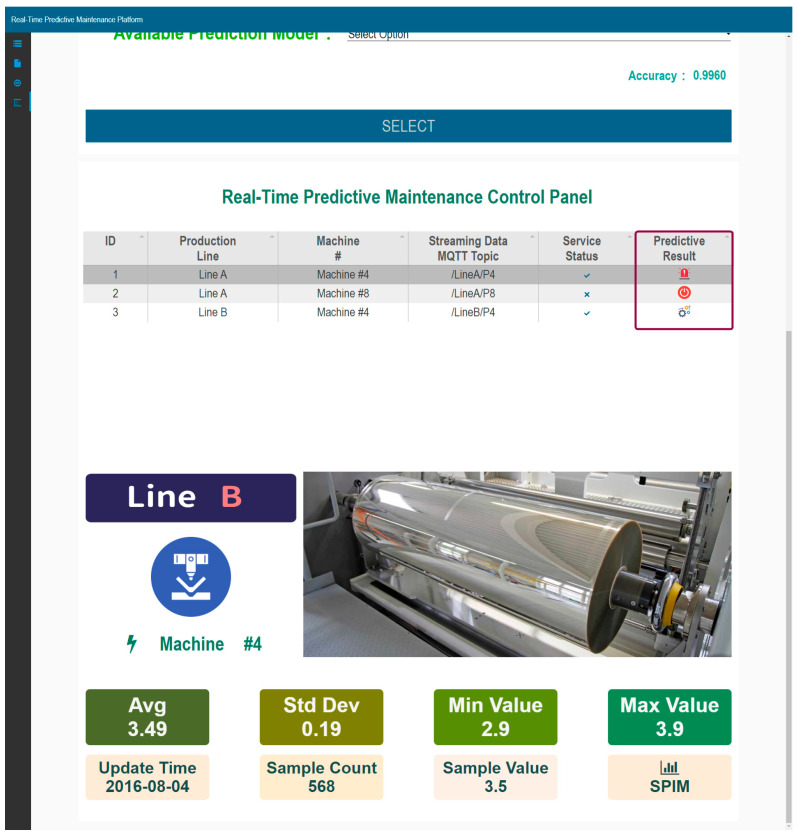
Model management and real-time predictive maintenance.

**Table 1 sensors-24-04237-t001:** Hardware and software configurations for Sparkling Water.

Hardware and Software Configurations	
CPU	8 Cores
Memory	32 GB
Operation System	Ubuntu 20.04
Python Version	3.8.10
Java Version	1.8.0_312
Scala Version	2.13.8
Spark Version	3.2.1
Hadoop Version	2.7
H_2_O Version	3.36.1.3 zumbo
Sparkling Water Version	3.36.1.3-1-3.2

## Data Availability

The data presented in this study are available on request from the corresponding author due to privacy.

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
