# Peer review of "Elevating Smart Manufacturing with a Unified Predictive Maintenance Platform: The Synergy between Data Warehousing, Apache Spark, and Machine Learning"

_sensors, 2024, doi:10.3390/s24134237_

Round 1

Reviewer 1 Report

Comments and Suggestions for Authors

Dear Authors,

The manuscript is inconsistent in content to a certain extent and does not even provide a general introduction to any scientific or even specific problem or an overview (survey) of technological developments. The focus changes from topic to topic.
The authors claim that the manuscript presents their own end-to-end platform, which is not clearly descibed.
The drawings (figures) look like paintings contain simple content with many IT product icons representing general information in a symbolic way. Some of them are borrowed from introductions from official documentation and technical guides for these open-source products. Diagrams do not need to contain these type of elements (icons), but may contain blocks with product names.

The literature list contains several outdated literature items.

If I can give you any advice, an article in this area should contain block diagrams of algorithms with defined operations (ETL/ELT) on data structures, such as: validation, cleaning, parsing, standardising, partitioning, bucketing, curating, extracting features, selecting features, encoding, training, evaluating, packaging, feature registering, containerising, deploying, serving, inferencing, triggering, ingesting, etc. You know what it refers to.

Sometimes the content looks like promotional or even a kind of marketing material, and the topic is too broad for the system and as the research topic for a single article.

I cannot recommend this manuscript for publication in a journal.

Comments on the Quality of English Language

Moderate editing of English language required

Author Response

Dear Reviewer

Thank you for your detailed and insightful feedback. We greatly appreciate the time and effort you have taken to review our manuscript. Your comments are invaluable to improving the clarity, focus, and overall quality of our work.

A point-by-point response to your comments is provided in the attachment. "Please see the attachment".

Reviewer 2 Report

Comments and Suggestions for Authors

The article constructed an end-to-end platform for fault detection in production machinery through the integration of a database module, a big data analysis module, and a real-time data processing engine. The database module consists of two parts: Extract, Transform, and Load (ETL) processes and Data Storage. ETL is used to collect sensor data from equipment. The big data analysis module consists of Feature Engineering and Model Generation. The Real-time Data Processing Engine Spark, performs inference and prediction. However, the article still has the following issues:

1.In section 3.1, Figure 6 mentions out-of-bounds cleaning and treatment of missing values. The treatment of missing data is a crucial aspect of the entire database system. Since the article involves real-time data processing, it is important to explain how the treatment of missing data ensures real-time data processing. It is recommended to supplement the explanation with the theoretical methods used for handling missing and noisy data in this paper.

2.In the big data analysis module, machine learning models are used, but the article lacks details on the training results and training strategies, such as data set partitioning and model training results.

3. The conclusion section lacks sufficient practical results to demonstrate that the developed platform can achieve predictive maintenance. Although the final paragraph of the conclusion clearly states that the platform has been successfully implemented, it would be beneficial to include successful application cases to support the current research.

4.Additionally, Figures 13 and 14 in this article are not very clear. It is suggested to replace them with clearer figures.

Author Response

Thank you for your detailed and insightful feedback. We greatly appreciate the time and effort you have taken to review our manuscript. Your comments are invaluable to improving the clarity, focus, and overall quality of our work.

A point-by-point response to your comments is provided in the attachment. "Please see the attachment".

Reviewer 3 Report

Comments and Suggestions for Authors

The article explores the integration of cutting-edge technologies to enhance predictive maintenance in smart manufacturing. The paper addresses the complexities of machinery in collaborative manufacturing environments and the risk of equipment failures. The research presents a sophisticated platform combining the organizational strengths of data warehousing with the processing power of Apache Spark, alongside the predictive capabilities of machine learning. This end-to-end system not only manages large volumes of time-series sensor data but also leverages these for real-time fault detection and operational optimization. The proposed model exemplifies significant innovation in smart manufacturing, offering improved operational reliability and proactive maintenance strategies that anticipate and mitigate potential machinery failures, thereby reducing costly downtimes.

The proposal is interesting, here are my comments:

1. How well does the proposed predictive maintenance model generalize across different types of manufacturing equipment and setups? This could be discussed in the introduction or at the beginning of the methodological model.

2. What measures are in place to ensure data privacy and security, given the extensive use of IoT devices and cloud technologies in the described predictive maintenance platform?

3. The authors could complement the state of the art related to Special Issue topics such as industry 5.0, with some of these references:

- Hassan, A., & Mhmood, A. H. (2021). Optimizing Network Performance, Automation, and Intelligent Decision-Making through Real-Time Big Data Analytics. International Journal of Responsible Artificial Intelligence, 11(8), 12-22.

- Cortés-Leal, A., Cárdenas, C., & Del-Valle-Soto, C. (2022). Maintenance 5.0: Towards a Worker-in-the-Loop Framework for Resilient Smart Manufacturing. Applied Sciences, 12(22), 11330.

4. How does the platform handle variations in data quality, especially given that IoT devices can vary widely in terms of the accuracy and reliability of the data they produce?

5. Could the authors elaborate on the real-time data processing capabilities of Apache Spark within this platform? Specifically, how does the system ensure minimal latency in fault detection when handling high-throughput data streams?

Author Response

Dear Reviewer,

Thank you for your insightful and constructive comments on our manuscript. We have carefully considered your feedback and have made significant revisions to address each of your points.

A point-by-point response to your comments is provided in the attachment. "Please see the attachment".

Round 2

Reviewer 1 Report

Comments and Suggestions for Authors

Dear Authors,

I have read the authors' response to my review. Thank you for answering.

But I still think that most of the figures are so simple that their content is too trivial. In my opinion, this is not at the level of a journal. I would say that the most inappropriate figure is fig. 3 and 6.
For example, fig. 3 shows an architecture using a data warehouse, at least it should include more of its basic details in the chosen context of the manuscipt:
- data sources can include: operational systems (databases), ERP, MES, CRP, CRM, flat files;
- data warehouse includes the following elements: raw data, meta data, summary data;
- there can be different data marts created with complex queries at the output from the data warehouse, or extracting processes for OLAP cubes, data files, data matrices;
- there can be also data pipelines at the output from the data warehouse for processing and modeling data to obtain results for: analysis, reports, data science, machine learning, business intelligence;

I could be said that there is nothing interesting in the manuscript for a researcher or even developer, because the authors do not present any algorithms (even as block diagrams) and some figures are based on the known diagrams (MongoDB cluster elements, Spark architectures, streaming elements) from software vendors (open-source and non-open-source) and uses just icons (instead of terms), and nothing is really known about industrial processes for maintenance predictions. It is impossible to be sure whether this is a real process and system. A manuscript in this form can serve as a kind of portfolio attachment rather than a scientific or survey article.

Comments on the Quality of English Language

English is fine.

Author Response

Comment: 

I have read the authors' response to my review. Thank you for answering.

But I still think that most of the figures are so simple that their content is too trivial. In my opinion, this is not at the level of a journal. I would say that the most inappropriate figure is fig. 3 and 6.

For example, fig. 3 shows an architecture using a data warehouse, at least it should include more of its basic details in the chosen context of the manuscript:

- data sources can include: operational systems (databases), ERP, MES, CRP, CRM, flat files;

- data warehouse includes the following elements: raw data, meta data, summary data;

- there can be different data marts created with complex queries at the output from the data warehouse, or extracting processes for OLAP cubes, data files, data matrices;

- there can be also data pipelines at the output from the data warehouse for processing and modeling data to obtain results for: analysis, reports, data science, machine learning, business intelligence;

I could be said that there is nothing interesting in the manuscript for a researcher or even developer, because the authors do not present any algorithms (even as block diagrams) and some figures are based on the known diagrams (MongoDB cluster elements, Spark architectures, streaming elements) from software vendors (open-source and non-open-source) and uses just icons (instead of terms), and nothing is really known about industrial processes for maintenance predictions. It is impossible to be sure whether this is a real process and system. A manuscript in this form can serve as a kind of portfolio attachment rather than a scientific or survey article.

Response:

Dear Reviewer,

Thank you for your continued feedback on our manuscript. We appreciate your insights and have taken them into serious consideration to further enhance the quality of our manuscript. Below, we address your specific comments regarding Figures 3 and 6 and provide details on the improvements made.

1) We have updated Figure 3 to include the recommended details, offering a more comprehensive representation of our data warehouse architecture as shown in the revised manuscript.

2) We concur that Figure 6 was not suitable and found it somewhat redundant. Consequently, we have removed this figure from the revised manuscript.

Reviewer 2 Report

Comments and Suggestions for Authors

No comment

Author Response

We sincerely appreciate the time and effort you have dedicated to reviewing our manuscript. Your feedback is invaluable in improving our work.

Reviewer 3 Report

Comments and Suggestions for Authors

The authors made my comments all right.

Author Response

(The authors gave the same response as above.)

Round 3

Reviewer 1 Report

Comments and Suggestions for Authors

Dear Authors,
Significant changes and extensions were introduced in the descriptions of methods and in the presentation of results. The manuscript provides a coherent research report, the quality of the figures have also been improved, and major shortcomings of the manuscript have been removed.
This version of the manuscript is clear and covers the research issues to an appropriate extent.
I can recommend the manuscript to be published.